# Stereoselective polar radical crossover for the functionalization of strained-ring systems

Florian Trauner[1,2], Rahma Ghazali[1], Jan Rettig [1], Christina M. Thiele [1] & Dorian Didier [1,2] ✉

Radical-polar crossover of organoborates is a poweful tool that enables the creation of two C-C bonds simultaneously. Small ring systems have become essential motifs in drug discovery and medicinal chemistry. However, step-economic methods for their selective functionalization remains scarce. Here we present a one-pot strategy that merges a simple preparation of strained organoboron species with the recently popularized polar radical crossover of borate derivatives to stereoselectively access tri-substituted azetidines, cyclobutanes and five-membered carbo- and heterocycles.

Strained carbo- and heterocycles have been brought to the forefront of medicinal chemistry and drug discovery programs in recent years as modulable sp³-rich 3D-isosters of diverse aromatic systems[1–4]. Besides exalting greater metabolic stability, it has been shown that small molecular scaffolds can help improve lipophilicity as well as pharmacokinetics[5–9]. The puckered conformation adopted by four-membered rings renders them ideal cores for drug discovery as they can balance both rigidity (observed in constraints systems such as propellanes[10–13] or cubanes[14,15]) and flexibility (conformers in larger cyclic scaffolds). Azetidines and cyclobutanes can, therefore, be used towards the three-dimensionalization of pyridyl and phenyl moieties, their substitution pattern following defined exit vectors (Fig. 1).

Recent step-economic strategies towards substituted azetidines include the work of Baran and Gianatassio on strain-release amination[16] and alkylation[17] of 1-azabicyclo[1.1.0]butanes (ABB) that provide an elegant route to 3-substituted structures, as well as our contribution on 1,3-bisarylations[18]. The group of Aggarwal reported the strain-release of ABB through boron-homologations for the synthesis of 3,3-bis functionalized azetidines[19–22]. Cyclobutanes were similarly obtained from metallated bicyclo[1.1.0]butanes[23–27]. Aside from strain-releasing strategies, substituted azetidines and cyclobutanes are traditionally approached through [2 + 2]-cycloadditions as recently illustrated by Schindler[28–31], Bach[32–35], Glorius[36–38] and Brown[39–43] as well as cyclizations and ring contraction and expansion reactions[44–50], which imply a pre-organization of the substituents around the structure of starting materials.

Aiming at the development of a synthetic toolbox that would allow diversely and selectively access functionalized four-membered building blocks, we set out to combine our expertise on the metalation of small heterocycles[51–58] and 1,2-boronate rearrangements[59–64] to design a simple one-pot sequence towards tri-substituted architectures. We envisioned that the inspiring work on polar radical crossover (PRC) pioneered by the groups of Studer[65,66], Aggarwal[67,68], Morken[69] and Renaud[70] could reveal fantastic opportunities to introduce three substituents at once on azetidines, cyclobutanes and other heterocycles, starting from corresponding cyclic alkenylmetal intermediates. To the best of our knowledge, control over the stereochemical outcome of this transformation remained moderate, as the radical process was only exemplified on acyclic alkenylboronates. However, for this strategy to take a consequent step further and enable a broad scope of applications, one would have to gain control over the spatial arrangement of vicinal substituents. We predicted that the diastereoselectivity of the 1,2-metalate rearrangement would be controlled thanks to the cyclic nature of our substrates, due to a locked configuration of reactive intermediates.

## Results and discussion
### Optimizations of reaction conditions
The first test was performed on azetinyllithium **1** (generated in situ), providing the bisorganoborinate **2** after the addition of $n$-BuBpin in THF. Generation of a radical species from nonafluorobutyl iodide under UV irradiation at −40 °C—as assumed from literature precedent[71]— provided the expected trisubstituted structure **3** in 44% with a moderate dr of 4:1 (Fig. 2, entry 1, table 1). We started optimizing the reaction parameters by assessing the importance of the stoichiometry of perfluorinated butyl-iodide on the yield. Under similar conditions, azetidine **3** was obtained with increased yields up to 78% with 1.5 equivalents of $C_4F_9I$ (entry 2), and comparable yield could be observed under blue light irradiation at −20 °C, keeping the same levels of diastereoselectivity. Solvent effects were examined next. While 1,3-dimethyl-2-imidazolidinone (DMI), dichloromethane and dichloroethane did not improve selectivity (entries 5–7), a diastereomeric ratio of 5:1 was measured in 2-methyl-THF (mTHF, entry 8).

It is interesting to note that the reaction performed in the absence of dodecane ($C_{12}H_{26}$) as standard only resulted in product formation with poor diastereoselectivity (entry 9, dr = 2:1).

[1]Technische Universität Darmstadt, Clemens-Schöpf-Insitut für Organische Chemie und Biochemie, Peter-Grünberg-Str. 4, 64287 Darmstadt, Germany.
[2]Ludwig-Maximilians Universität, Department Chemie, Butenandtstr. 5, 81377 München, Germany. ✉e-mail: dorian.didier@tu-darmstadt.de

**Fig. 1 | Previous work and present contribution on radical-polar crossover reactions.** Achievements reached in the diastereoselective radical-polar crossover of cyclic systems such as 4- and 5-membered carbo- and heterocycles, in contrast with previous work by Studer, Aggarwal, Morken and Renaud on acyclic systems.

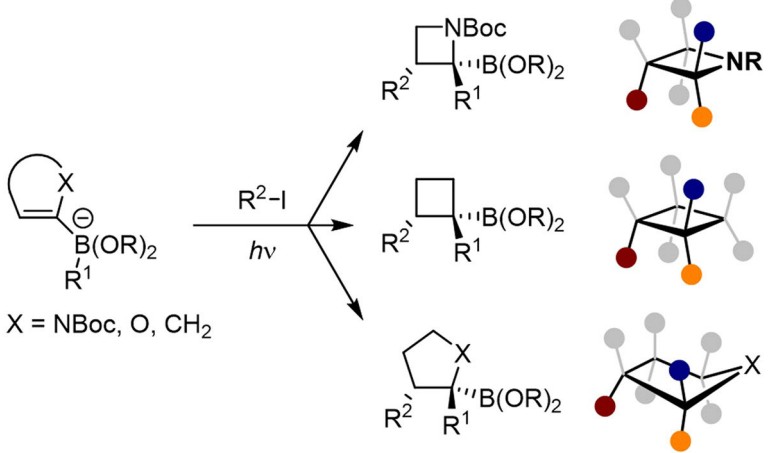

With these encouraging results in hands, the influence of steric effects was evaluated by changing the ligand structure on the boron atom. The process was reiterated employing organoboron species **A**–**H** under the conditions displayed in entry 8. While reagents **A**, **B** and **C** gave similarly high yields (63–93%), the groups present on the pinacol scaffold allowed for a broad modulation of dr values, a maximum being reached for reagent **A** ($n$-BuB$^E$pin, dr = 8:1). Only traces of products were observed with 1,3-propyldiols (**E** and **F**) and the isopropyl-pinacol structure **D**. Surprisingly low dr were obtained using phenyl-pinacol derivative **G** or "Bmac" **H**[72], products being additionally isolated only in moderate yields.

With a fair adequation between conversion and diastereoselectivity, ethyl-pinacol ($^E$pin) ligands **A** were further employed to explore the scope of the reaction. In addition to positively influencing the dr, products obtained with ligand **A** showed high stability on silica (avoids protodenoronation) when compared to classical pinacols.

### From azetines to trisubstituted azetidines

The scope of the transformation was first assessed on azetinyllithium species **1**, in situ generated from 3-methoxyazetidines.[15f] Coordination to an organoboron derivative R$^1$-B$^E$pin primarily gives borinate **2**, which was then engaged in PRC after the solvent switch to mTHF and addition of the radical precursor R$^2$-I under blue light irradiation (Fig. 3). Products **3a**–**e** were synthesized in moderate to good yields from alkylboron reagents and per-fluorinated radical precursors with good dr values (8:1–20:1), except for methylboronic ester (**3e**, dr = 1:1), which might come from a lack of steric effects (vide infra).

Excellent dr (>20:1) was observed when employing ethyl 2,2-difluoro-2-iodoacetate (**3f**, **g**), and we noticed a general trend for the iodoacetates to give increased dr (**3k**–**m**, dr > 20:1) in comparison with other radical precursors (**3h**–**j**, up to 7:1 dr). The lack of sterical hindrance in the case of methyl boronic ester (**3n**, dr = 2:1). Arylboronic ester also tended to decrease the selectivity (**3i** and **3o**, 2:1 to 5:1 dr). In all cases, the 1,2-metallate rearrangement proceeded in a trans-selective fashion (R$^1$ vs. R$^2$), as supported by thorough analytic measurements and experimental data (vide infra). Furthermore, substituted azetidines proved to be stable under basic conditions, and we were able to hydrolyze the ester moiety into the corresponding carboxylic acid **3l'** in good yield (76%), keeping the dr value above 20:1.

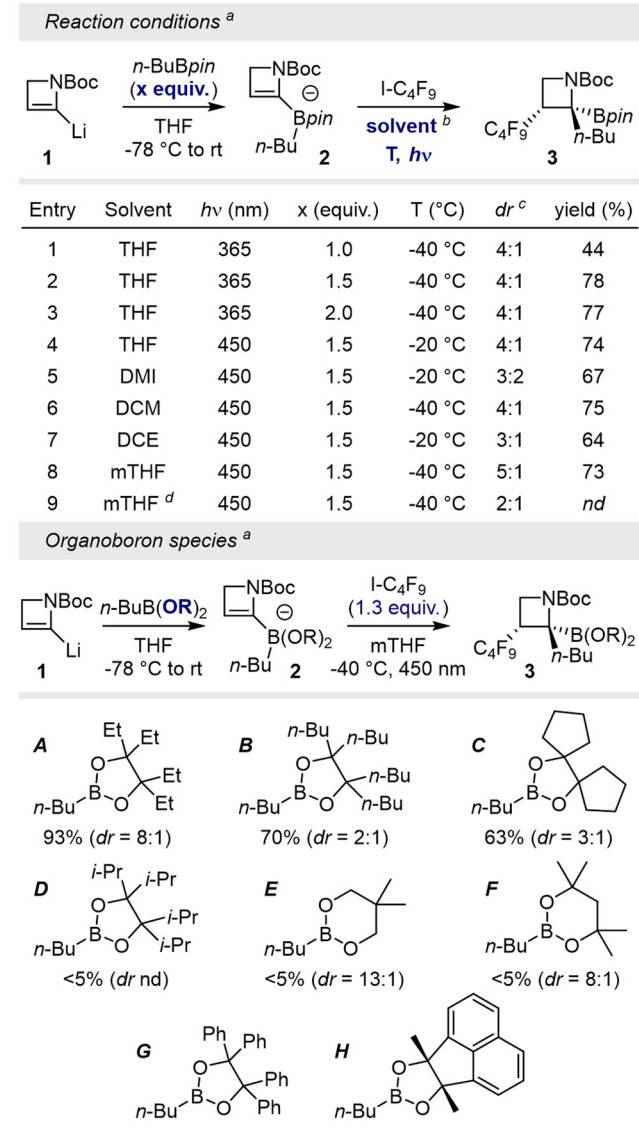

**Fig. 2 | Optimizations of the polar radical crossover on in situ generated azetinyllithium species. a** Indicated yields have been assessed by GC-analysis of the crude mixture; $C_{12}H_{26}$ (1 vol%, 30 mL) was used as standard. b for reactions performed in other solvents than THF, a solvent switch was performed after the removal of THF for the crude mixture. c indicated dr were measured on the crude mixture by 19F-NMR. d The reaction was performed in the absence of C12 standard.

**Fig. 3 | Evaluation of the PRC sequence scope for the synthesis of trisubstituted azetidines from azetinyllithium species.** Azetinyllithium **1** is generated in situ by α-lithiation/elimination/α-lithiation, starting from commercially available 3-methoxyazetidines. The first steps are conducted in THF, but the best distereoselectivities are achieved by performing the last rearrangement step in Me-THF.

### From cyclobutenes to trisubstituted cyclobutanes

The stereoselective synthesis of cyclobutanes through PRC was envisioned next from readily available starting cyclobutenylboronic esters **4** (Fig. 4), applying previously optimized conditions (Fig. 2). The scope of the transformation was explored by varying both radical precursors (perfluorinated alkyl iodides and iodoacetates) and nature of the organolithium species (alkyl- and aryl-lithium). A range of functionalized cyclobutanes (**6a–f**) was isolated in good yields and excellent dr (up to 20:1) with the exception of cyclopropylboronic ester (**6c**, dr = 6:1). Interestingly, an iodomethylketone proved to be an efficient radical precursor for this reaction (**6e**, dr = 18:1), as well as unprotected iodoacetamide (**6f**), although with slightly decreased diastereoselectivity (dr = 7:1). It is important to note that the diastereoselectivity of the metallate rearrangement step on cyclobutyl-intermediates is generally superior to the one on azetidinyl-species.

Given the high levels of diastereoselectivity observed in cyclobutenylboron species, two derivatives possessing the thiophene sub-unit typically found in Canagliflozin (a drug used in the treatment of type 2 diabetes)[73] were synthesized, employing iodoacetate (**6g**) and trifluoromethyliodide (**6h**).

### Scope of the transformation on cyclopentenes

Larger carbocyclic systems were explored next, starting from stable, storable cyclopentenylboronic ester **7a** (Fig. 5). Coordination of an organolithium reagent ($R^1$-Li) to **7** promotes the formation of the bisorganoborinate **8**. Radical crossover was further initiated under blue light irradiation at −40 °C after the solvent switch and addition of iodoacetates as radical precursors. Trisubstituted cyclopentanes **9a–e** were obtained with high yields and stereochemical ratio (up to 20:1 dr), except for the addition of aryllithium species (**9d**, dr = 2:1), as previously observed. While switching the radical precursor for a *tert*-butyl ester gave similar results, both in efficiency and

**Fig. 4 | Application of the PRC sequence to the tris-functionalization of cyclobutanes from isolated cyclobutenylboronic ester derivatives 4.** The scope of the reaction (R¹) is evaluated for both aryl and alkyl substituents, showing a generally higher trend in diastereoselectivities than for azetidines.

**Fig. 5 | Generalization of the PRC sequence to larger rings.** Evaluation of the scope for the diastereoselective formation of tris-functionalized cyclopentanes from preformed cyclopentenylboronic esters 7. Steroide-derivatives are also described, although with a lower diastereomeric ratio.

**Fig. 6 | Stereoselective functionalization of pyrrolidines, THF (top) and norbornanes (bottom) via PRC.** Evaluation of the reaction scope using readily accessible lithiated dihydropyrroles and dihydrofuranes **10**, as well as norbornenylboronic esters **13**.

diastereoselectivity (**9a'** and **9b"**), trifluoromethyl iodide furnished product **9e** with slightly lower levels of selectivity (7:1 dr).

These compounds also proved stable under basic conditions, product **9b** being hydrolyzed into **9b'** in excellent yield (91%). Interestingly, we demonstrated the applicability of this stereoselective method to the functionalization of estron scaffolds. The stable alkenylboronic ester substrate **7b** was first accessed in few steps from (+)-estrone 3-methyl ether and further engaged in PRC, leading to products **9f** and **9g** in high yields, although the diastereoselectivity could only reach 3:1. It is, however, important to note that addition of the in situ generated radical species occurred on the least hindered face of the cyclopentenylboronates (*cis* to the methyl substituent).

**Scope of the transformation on heterocycles and norbornenes**
Although heterocyclic five-membered rings were efficiently engaged in PRC to provide trisubstituted pyrrolidine **12a** and THF **12b, c** (up to 96% yield), the diastereomeric ratio could only reach up to 3:1 (Fig. 6).

Norbornenylboronic ester **13** was readily prepared in a few steps from commercially available norbornene and proved to be a suitable building block for PRC. It efficiently provided substituted structures **15a–c** in both

high yields (72–92%) and stereoselectivities (dr > 20:1). The formation of the major diastereoisomer can be explained by the addition of the radical species on the least hindered bridged side of the norbornenyl-substrate, followed by an antiperiplanar 1,2-metallate rearrangement.

**Determination of the relative configuration**
Stereochemical relationships between newly introduced substituents were studied first by NOE and HOE on compounds **3e**, **3d** and **3a** (Fig. 7, see supporting information). The poor diastereomeric ratio (ca. 1:1) obtained for **3e** allowed us to separate both diastereoisomers (**3e**$_{syn}$ and **3e**$_{anti}$) in sufficient quantities for NMR experiments. In the case of **3e**$_{anti}$, while a strong NOE was observed between the H-atom at position 3 and the methyl group at position 2, we could not detect any significant HOE ($^1$H–$^{19}$F), supporting the anti-configuration of R$^1$ and R$^2$ groups. Inversed observations were made for compound **3e**$_{syn}$, for which a very weak NOE was detected, but with a strong HOE ($^1$H–$^{19}$F) between the perfluorinated chain at position 3 and the methyl group at position 2. Analogy was then made for compound **3d**, isolated with a higher dr (>20:1), for which we assigned the anti-configuration through both detections of a strong H–CH$_3$ NOE and the

**Fig. 7 | Nuclear Overhauser effect (NOE) and heteronuclear Overhauser effect (HOE) on synthesized azetidines to support the assigned relative configuration of the three substituents (top).** Experimental assessment of the relative configuration through oxidative C–B bond cleavage followed by lactonization (middle). Proposed model for the stereoselectivity and elements of support for the observed relative configuration (bottom).

absence of significant HOE ($^1$H-$^{19}$F). Similarly, strong NOE on compound **3a** (dr = 8:1) allowed us to assign its anti-configuration.

Brown oxidation (NaOH, H$_2$O$_2$) on **9b** led to the bicyclic compound *cis*-**16** by subsequent lactonization, which brought additional support to the assigned stereochemistry. We propose that the R$^2$-chain introduced from the radical precursor shields one of the two diastereotopic faces of the cyclic intermediate, disfavoring the 1,2-metallate rearrangement of R$^1$ from the same face [TS1], to favor an antiperiplanar addition [TS2].

## Further applications

Finally, we evaluated the robustness and the configurational stability of our cyclic organoboron systems under different conditions (Fig. 8). Switching the Boc protecting group on the nitrogen atom (**3a**) for a benzyl group in a two-step sequence led to *N*-benzyl product **17a**, which was isolated in 67% yield with retained diastereomeric ratio (8:1). **17b** was obtained through the same reaction sequence from its NBoc derivative parent in 56%, without the need for intermediate purification. Ligand exchange on **17b** also proceeded with stereoretention towards the potassium trifluoroborate salt **19** in good yields (75%). Using a modified Brown oxidation sequence in which the B$^E$pin group was transiently transformed into the more reactive BCl$_2$

derivative, the bicyclic azetidine-based aminoacetal **20** was isolated in 84% with retention of the stereochemistry (dr = 9:1). A similar procedure than on the azetidines was used to stereoretentively transform the cyclopentyl-compound **9c** into the corresponding trifluoroborate salt **21** with high efficiency (85%). Surprisingly, when compound **9a'** was treated with BCl$_3$, bicyclic oxaborinanone **22** was obtained upon the addition of water and concentration in vacuo, supporting once again the relative stereochemistry observed for polar radical crossover on cyclic systems.

## Conclusion

We have developed a robust, efficient and highly diastereoselective sequence based on polar radical crossover that allows to access stereodefined trisubstituted azetidines, cyclobutanes, cyclopentanes, THFs and pyrrolidines. Fine tuning of reaction conditions revealed the importance of diol ligands on the boron to maximize the stereoselectivity, opening thereof new opportunities in boron-based synthetic methodologies.

## Methods

See Supplementary methods.

See Supplementary Data 1.

**Fig. 8 | Post-functionalization of cyclic boronic esters.** Including manipulation of N-protecting groups, ligand exchanges on the boron atom and oxidation / lactonization on azetidines, ligand exchange and oxidative formation of oxaborinanone on borylated cyclopentanes.

## Data availability

Data for this manuscript has been deposited in figshare: https://doi.org/10.6084/m9.figshare.25907506. "Supplementary Methods" contains detailed protocols for the preparation of substrates, reaction optimizations, scope evaluation and description of analytical data ($^1$H and $^{13}$C NMR, HRMS). "Supplementary Data 1" contains all $^1$H and $^{13}$C NMR spectra.

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

## Acknowledgements
D.D., F.T., C.M.T., and J.R. are grateful to the Deutsche Forschungsgemeinschaft (Heisenberg fellowship: DI 2227/4-1), to the Ludwig-Maximilians University (LMU Excellence) and the Technical University of Darmstadt for Ph.D. funding and financial support. R.G. thanks the ERASMUS program.

## Author contributions
Conceptualization and supervision: D.D. Experimental investigation: F.T., R.G. Structure analysis: J.R., C.M.T. Writing: D.D., C.M.T. Supporting information: F.T., R.G., J.R. Proofreading: D.D., C.M.T., F.T., R.G., and J.R.

## Competing interests
The authors declare no competing interests.
