## [Peer Review File · Communications Chemistry]

Reviewers' comments:

Reviewer #1 (Remarks to the Author):

Didier reports on the functionalization of strained rings. The reaction works by formation of a boronate followed by polar Radical cross over 1,2-shift. The reaction is based on work from Studer and Aggarwal, among others. The reaction conditions are not too different from prior work.

The synthesis of strained rings with control of stereochemistry is of high importance to the drug discovery industry. This method directly addresses this problem and generates useful products. The conditions are simple, and the scope is quite broad. Primarily for the latter reason, I think this work is suitable for publication.

However, I think the authors need to do more with functionalization of the Bepin part. This is especially true for the n-heterocycles.

Scheme 5 is the same as Scheme 6. I think Scheme 6 should have pyrrolidines, THF and norbornenes.

What are some key limitations?

I don't understand the connection to Canagliflozin. The sugar was undoubtedly important. Why is replacing with a azetidine important. If the authors are trying to show tolerance to thiophenes, just show that. There is no need to connect to Canagliflozin

Reviewer #2 (Remarks to the Author):

Trauner and Didier et al. have outlined a stereoselective and step-economic method for generating functionalized strained carbo- and heterocycles. This approach involves utilizing organoboron and alkyl halide as radical precursor. While previous studies have explored the polar radical crossover strategy for functionalizing vinyl boron ate complexes followed by 1,2-migration, this report takes it a step further by extending the method to produce highly functionalized strained molecules like azetidine and cyclobutene. The process demonstrates good diastereoselectivity under mild and economically feasible conditions. The article highlights the broad applicability of this methodology and also provides support through mechanistic studies. Despite the importance of the reported method, a significant revision is necessary for the article to be considered for publication. Several concerns have been raised:

1. Scheme 5 and Scheme 6 are identical in the manuscript, resulting in the omission of numerous structures discussed in the text. Such errors are unacceptable in a research manuscript submission. Please rectify this issue.
2. The mention of "X equiv." appears in the wrong place in the optimization table. It is assumed that the author is referring to alkyl halide equiv. (step 2nd) screening in the table, not boronic ester (1st step) equiv.
3. It is of interest whether the author conducted any studies to investigate the impact of dodecane (used as an internal standard) on improving diastereoselectivity. Was there any co-solvent effect considered?
4. Was the diastereoselectivity in THF solvent tested when Ethyl-Bpin (A) was used? This entry is not

present in the optimization table. If a similar diastereomeric ratio profile is observed, the solvent-switching process may be avoidable.

Point-by-point answer to reviewers comments:

Reviewer #1:

- 1. “The synthesis of strained rings with control of stereochemistry is of high importance to the drug discovery industry. This method directly addresses this problem and generates useful products. The conditions are simple, and the scope is quite broad. Primarily for the latter reason, I think this work is suitable for publication.”**

We thank the reviewer for his encouraging comment.

- 2. “However, I think the authors need to do more with functionalization of the Bepin part. This is especially true for the n-heterocycles.”**

The post-functionalization of boron derivatives that were obtained during this study is of course of great interest, as we are working actively on the sophistication of strained carbo- and heterocycles. We are still working on different methods to do so, including C-C bond formation. However, these methods still need optimizations. Moreover, we chose to fully focus on the polar-radical crossover sequence rather than filling up the manuscript with post-transformations. This being said, we agree that proof of functionalization should be demonstrated, especially on azetidines, and we added 2 transformations to scheme 8: 1) The transformation of 17b into the corresponding BF₃K salt 19, given that the Boc group be replaced by a benzyl group. 2) The more interesting lactonization procedure used to support the relative stereochemistry was employed on 18 to provide the NBn-azetidine-based aminoacetal compound 20 in 84% yield. Importantly, these two reactions proceeded with full retention of the stereochemistry. 18 was synthesized from 3p (previously not described in Scheme 3), which has been added to both manuscript and supporting information.

- 3. “Scheme 5 is the same as Scheme 6. I think Scheme 6 should have pyrrolidines, THF and norbornenes.”**

This is a regrettable mistake. Scheme 6 has been correctly displayed in the new version of the manuscript.

- 4. “What are some key limitations?”**

Aside from some examples leading to lower yields, there are no limitations in terms of reactivity. Our initial program was to assess the feasibility of the sequence on strained ring systems. However, as the project developed, we went on with studying larger ring systems. For 6- and 7-membered rings, although high yields are obtained, a large decrease in diastereoselectivity is observed. We did not describe those systems in the manuscript, but this could – if requested – be added as a footnote.

- 5. “I don’t understand the connection to Canagliflozin. The sugar was undoubtedly important. Why is replacing with an azetidine important. If the authors are trying to show tolerance to thiophenes, just show that. There is no need to connect to Canagliflozin.”**

We understand. The structure of Canagliflozin was removed from Scheme 4, and the text adapted to mention the tolerance towards thiophenes.

Reviewer #2 (Remarks to the Author):

1. **“Trauner and Didier et al. have outlined a stereoselective and step-economic method for generating functionalized strained carbo- and heterocycles. This approach involves utilizing organoboron and alkyl halide as radical precursor. While previous studies have explored the polar radical crossover strategy for functionalizing vinyl boron ate complexes followed by 1,2-migration, this report takes it a step further by extending the method to produce highly functionalized strained molecules like azetidine and cyclobutene. The process demonstrates good diastereoselectivity under mild and economically feasible conditions. The article highlights the broad applicability of this methodology and also provides support through mechanistic studies.”**

We thank the reviewer for this encouraging note.

2. **“Scheme 5 and Scheme 6 are identical in the manuscript, resulting in the omission of numerous structures discussed in the text. Such errors are unacceptable in a research manuscript submission. Please rectify this issue.”**

As for reviewer #1, this is a regrettable mistake, which was corrected in the new version of the manuscript.

3. **“The mention of "X equiv." appears in the wrong place in the optimization table. It is assumed that the author is referring to alkyl halide equiv. (step 2nd) screening in the table, not boronic ester (1st step) equiv.”**

In Scheme 2, X equiv. represents the equivalent amount of organoboron derivative used in the sequence.

4. **“It is of interest whether the author conducted any studies to investigate the impact of dodecane (used as an internal standard) on improving diastereoselectivity. Was there any co-solvent effect considered?”**

Indeed, this is a question we asked ourselves when conducting the optimizations. We also considered co-solvent effects. For this reason, we conducted tests with different amounts of dodecane (1%, 5% and 10%). However, while 1% (dr = 7:1, 92%) and 10% (dr = 8:1, 88%) showed increased diastereoselectivities in comparison with the absence of dodecane (dr = 6:1, 93%), 5% gave higher dr values (dr = 8:1) and yields (93%). The study is fully described in the supporting information.

5. **“Was the diastereoselectivity in THF solvent tested when Ethyl-Bpin (A) was used? This entry is not present in the optimization table. If a similar diastereomeric ratio profile is observed, the solvent-switching process may be avoidable.”**

This was tested as suggested, resulting in the products being obtained with drastically lower yields as well as diastereoselectivity. A comment was added to the supporting information.

REVIEWERS' COMMENTS:

Reviewer #1 (Remarks to the Author):

The authors have addressed my concerns. I find this suitable for publication

Reviewer #2 (Remarks to the Author):

I am happy and satisfied with the response letter. I recommend this article for publication now.